| Open Peer Review | Clinical Microbiology | New-Data Letter

# Detection of varicella-zoster virus in saliva and plasma of patients with typical and atypical herpes zoster

Yinling Li,[1] Xiuli Zhao,[1] Jiewei Chen,[1] Zhen Pu,[1] Chiyu Zhang,[2] Aizhen Zhou,[1] Zhenzhou Wan[1]

**KEYWORDS** typical herpes zoster, atypical herpes zoster, saliva, varicella-zoster virus, plasma

Herpes zoster (HZ), a common skin disease caused by varicella-zoster virus (VZV) infection, is typically characterized by unilateral erythematous rash overlaid with clustered vesicles and varying degrees of neurological symptoms (1). Postherpetic neuralgia (PHN) is the most common sequela of HZ. Approximately 75% of PHN cases result from delayed diagnosis and treatment during the acute phase, and this proportion is increasing annually (2, 3). In addition to typical herpes zoster, there are apparently various types of herpes zoster, including incomplete herpes zoster, zoster sine herpete, disseminated zoster, generalized zoster, herpes zoster ophthalmicus, Ramsay Hunt syndrome, visceral herpes zoster, and so on. Antiviral therapy can significantly decrease the duration and severity of HZ-associated pain (4, 5). Missed diagnosis and/or misdiagnosis of atypical HZ not only increases the healthcare costs and psychological burden on patients but also delays treatment and raises the risk of complications and PHN. The diagnosis of typical HZ is often based on clinical symptoms; however, the diagnosis of atypical herpes zoster proves challenging. Furthermore, because of the similarities in the clinical presentations of VZV infection with herpes simplex virus infection, as well as other diseases (e.g., erythema multiforme) (6), the accurate diagnosis of VZV infection is of clinical significance for the initiation of appropriate antiviral treatment.

Previous studies have shown that VZV DNA can be detected in the blood, saliva, bronchoalveolar lavages, cerebrospinal fluids, and even urine at the time of HZ onset (7, 8). In particular, Guiraud et al. reported that VZV DNA can be detected in the blood samples of 77% of immunocompromised patients before herpes zoster rash onset, and 100% of patients with typical HZ symptoms (9). Furthermore, Quinlivan et al. reported a 100% positive rate of VZV DNA in the blood during the acute phase of HZ, and VZV DNA remained to be detected in 91% of them at 6 months after the symptom onset (10). Compared with blood specimens, saliva has the advantage of a non-invasive, easy, and cost-effective collection. Choi et al. reported a 72% positive rate of VZV DNA in the saliva samples of patients with Ramsay Hunt syndrome (11). Park et al. reported that the detection rate of VZV DNA was 88% in saliva, but only 28% in the plasma samples of patients with herpes zoster (12). VZV is thought to be bound to cells in the blood, and its level in plasma is almost half that in the whole blood (13), which can explain the lower positive rate of VZV DNA in plasma than in the whole blood. However, the presence of VZV DNA in the saliva and plasma samples of patients with atypical HZ is rarely of concern.

Here, we performed a prospective study to detect and compare the presence of VZV DNA in plasma, saliva, and/or blister fluids from patients with typical (vesicular) and atypical (non-vesicular) HZ using our previously described quantitative polymerase chain reaction (qPCR) assay (14). Both the vesicular and non-vesicular groups contained 35 patients each, with 22 males and 13 females in the former and 15 males and 20

Address correspondence to Aizhen Zhou, 13905262835@163.com, or Zhenzhou Wan, wanlv@126.com.

The authors declare no conflict of interest.

females in the latter (Table 1). All patients enrolled in this stu2dy were diagnosed by the dermatologists (i.e., authors: Yinling Li, Jiewei Chen, Zhen Pu, and Aizhen Zhou) of Taizhou Fourth People's Hospital. The atypical HZ contained incomplete herpes zoster, zoster sine herpete, and disseminated zoster. The time from disease onset was relatively shorter in the vesicular group than in the non-vesicular group ($P = 0.08$). There was no significant difference in gender and age between both groups ($P > 0.05$). Furthermore, saliva was also collected from 40 staff at Taizhou Fourth People's Hospital who have no major underlying diseases as healthy controls. All these staff volunteers gave their samples freely and without duress.

Viral DNA was extracted from all samples using a qEx-DNA/RNA Viral Extraction Kit (Tianlong, Xi'an, China) and subjected to the qPCR assay. No VZV DNA was detected in the saliva of healthy people. VZV DNA was detected in all blister fluids (100%) of 35 patients with typical HZ, supporting the fact that these patients were all infected with VZV (Table 1). No VZV DNA was detected in the plasma samples, regardless of the vesicular or non-vesicular groups, supporting the previous suggestion that the plasma might not be the best sample for diagnosis of VZV infection (12). Salivary VZV DNA was detected in seven (20%) patients with atypical HZ and in 14 patients (40%) with typical HZ (Table 1). It is not surprising that blister fluid had the highest positive rate (100%) of VZV DNA, significantly higher than saliva and plasma ($P < 0.0001$ for both). The higher positive rate of VZV DNA in saliva than in plasma supported the previous observation of Park et al. (12) and suggested that saliva is more suitable for VZV detection than plasma. In addition, the detection rate of salivary VZV DNA in this study was relatively lower than those in the previous studies (12), which might be partly ascribed to different methodologies, including DNA extraction and the qPCR assay. Although the positive rate of salivary VZV DNA in patients with typical HZ was twice that in patients with atypical HZ, the difference does not reach a statistically significant level ($P = 0.068$). We further compared the threshold cycle (Ct) values of VZV DNA by the qPCR assay in different groups. There was no significant difference in the Ct values of the saliva samples between the atypical and typical HZ groups, while significantly lower Ct values were

**TABLE 1** Detection of varicella-zoster virus (VZV) DNA in the plasma, saliva, and blister fluid of patients with typical and atypical herpes zoster[a]

| Items | Atypical herpes zoster (non-vesicular) | Typical herpes zoster (vesicular) | P value |
|---|---|---|---|
| Number of cases[b] | 35 | 35 | NA |
| Average age (range: years old) | 58.6 (9–75) | 63.3 (39–78) | 0.132 |
| Mean time from onset (range: days) | 5.1 (2–10) | 4.1 (1–10) | 0.080 |
| Gender | | | |
| Male | 15 | 22 | 0.094 |
| Female | 20 | 13 | |
| VZV DNA | | | |
| Plasma | | | |
| Positive | 0 | 0 | 1.000 |
| Negative | 35 | 35 | |
| Saliva | | | |
| Positive | 7 | 14 | 0.068 |
| Negative | 28 | 21 | |
| Blister fluid | | | |
| Positive | NA[c] | 35 | NA |
| Negative | NA | 0 | |

[a]Comparison was performed using $t$-test and $\chi$ test with SPSS 26.0.
[b]All patients were immunocompetent at least during this study. In addition, there were two patients who had received surgery to remove cancer with and without subsequent chemotherapy about 2 years before sample collection.
[c]NA: Not Applicable.

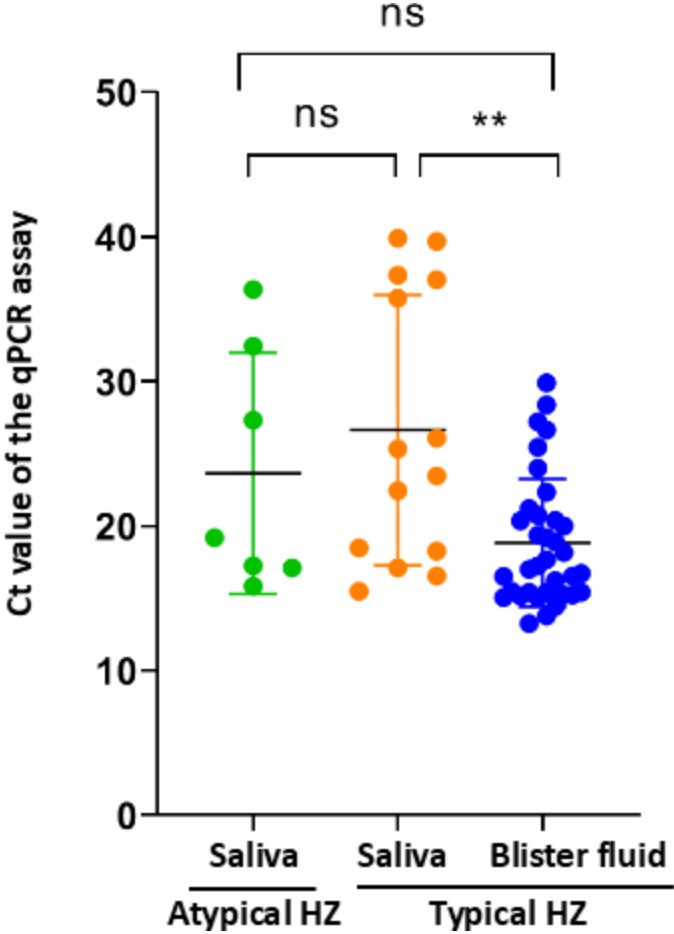

**FIG 1** Comparison of the Ct values of VZV DNA by the quantitative polymerase chain reaction (qPCR) assay in different samples. ns, not significant. **, $P < 0.01$.

observed in the blister fluid of the typical HZ group (Fig. 1), indicating a higher VZV DNA load in the blister fluid.

There are two limitations. First, although VZV DNA is often detected in whole blood, we did not test the whole blood samples. So, we were unable to compare the detection rates of VZV DNA between blood and saliva. Second, despite the feasibility of the qPCR method to various clinical samples, we did not previously validate the VZV qPCR assay using saliva samples.

In conclusion, we report that VZV DNA can be 100% detected in the blister fluid of patients with typical HZ, indicating that blister fluid can be used in distinguishing VZV and HSV infections that share similar clinical symptoms (6). VZV DNA can be detected in the saliva of some patients with typical and atypical HZ, implying that saliva can be used as an easy and cost-effective approach to assisting the clinical diagnosis of atypical HZ. In view of the non-invasive and easy collection of saliva, the development of simple, rapid, and highly sensitive point-of-care testing for VZV DNA is encouraged.

## ACKNOWLEDGMENTS

We thank Lin Wu for his help in this research.

This research was funded by the Science and Technology Support Foundations of Taizhou (No. TSZD-202201) to Y. L.

Z.W. and C.Z. conceived and designed the study. Y.L. and X.Z. performed the experiments. Y.L., J.C., Z.P., and A.Z. collected samples. Y.L., Z.W., and C.Z. analyzed the

data. Y.L. wrote the manuscript. C.Z. and Z.W. revised the article. Z.W. and A.Z. supervised this study. All authors read and approved the final version of the manuscript.

## AUTHOR AFFILIATIONS

[1]Taizhou Fourth People's Hospital, Taizhou, Jiangsu, China
[2]Shanghai Public Health Clinical Center, Fudan University, Shanghai, China

## AUTHOR ORCIDs

Chiyu Zhang http://orcid.org/0000-0001-8735-9857
Aizhen Zhou http://orcid.org/0009-0006-2001-9026
Zhenzhou Wan http://orcid.org/0000-0001-9624-4226

## ETHICS APPROVAL

The use of clinical samples was conducted in accordance with the Declaration of Helsinki and approved by the Ethics Committee of Taizhou Fourth People's Hospital (No. 2023-EC/TZFH-0025). Oral informed consents were obtained from the patients recruited in the study.

## ADDITIONAL FILES

The following material is available online.

Open Peer Review

**PEER REVIEW HISTORY (review-history.pdf).** An accounting of the reviewer comments and feedback.

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
