## [Reviewer comments · Microbiology Spectrum]

Microbiology Spectrum

Detection of varicella-zoster virus in saliva and plasma of patients with typical and atypical herpes zoster

Yinling Li, Xiuli Zhao, Jiewei Chen, Zhen Pu, Chiyu Zhang, Aizhen Zhou, and Zhenzhou Wan

Corresponding Author(s): Zhenzhou Wan, Taizhou Fourth People's Hospital

Review Timeline:

Submission Date:	June 25, 2024
Editorial Decision:	July 25, 2024
Revision Received:	August 13, 2024
Editorial Decision:	August 21, 2024
Revision Received:	August 23, 2024
Accepted:	August 23, 2024

Editor: Paschalis Vergidis

Reviewer(s): Disclosure of reviewer identity is with reference to reviewer comments included in decision letter(s). The following individuals involved in review of your submission have agreed to reveal their identity: Bismarck Steven Bisono-Garcia (Reviewer #1); Hong Du (Reviewer #3)

Transaction Report:

DOI: <https://doi.org/10.1128/spectrum.01549-24>

Re: Spectrum01549-24 (Detection of varicella-zoster virus in blood and saliva of patients with typical and atypical herpes zoster)

Dear Dr. Zhenzhou Wan:

Thank you for the privilege of reviewing your work. Below you will find my comments, instructions from the Spectrum editorial office, and the reviewer comments.

I agree with the reviewers' comments about providing clinical information, particularly whether patients were immunocompromised. Please see attachment for comments of Reviewer 2.

Revision Guidelines

Sincerely,
Paschalis Vergidis
Editor
Microbiology Spectrum

Reviewer #1 (Comments for the Author):

In the small methods paragraph please specify statistical methods applied. Lack of description on how non vesicular zoster was diagnosed and what specifics the patients had, especially how the difference from HSV was made.

Reviewer #2 (Comments for the Author):

The manuscript is clear and well-written, statistical analysis is well done and presented results are of interest for physicians who could deal with suspicion of atypical herpes zoster. Some clinical details however are needed to further characterise the study population and the assay that was used to support conclusions. Please see attachment for detailed comments.

Reviewer #3 (Comments for the Author):

In this paper, Li et al detected and compared the VZV DNA among blood and saliva samples from patients with typical and atypical herpes zoster. The results have clinical implication for early diagnosis of atypical herpes zoster, as well as distinguishing VZV infection from HSV infection, and deserve publication.

Comments:

- 1.The definition of atypical herpes zoster is not completely accurate. Atypical herpes zoster also contains disseminated herpes zoster and visceral herpes zoster, etc. Furthermore, some cases without skin lesions have pain.
- 2.The clinical implication and/or importance of accurate diagnosis of VZV infection should be discussed. For example, how to distinguish atypical herpes zoster from other visceral diseases, and to distinguish typical and atypical herpes zoster from herpes simplex when it occurs in the skin and mucosa.
- 3.More clinical information of the patients should be provided. For example, did the patients receive antiviral treatment before sample collection?
- 4.The viral load or at least the CT values of the qPCR assay can be analyzed and compared between different groups.

Review :

Detection of varicella-zoster virus in blood and saliva of patients with typical and atypical herpes zoster

Overall comment: The paper is clear, well-written and easy to read. I hope it will get published. Results from blood positivity are a bit surprising from my experience but could be explained by different patient populations and different diagnosis methods.

Line 15-16: I don't agree with the assumption that early treatment decrease PHN. The reference 2 only addresses median time to resolution of the acute symptoms, not PHN. Their figure 1 for instance is extremely relevant: median time to resolution is shorter but proportion of patients with ZAP at 24 weeks is roughly similar. There is also a Cochran meta-analysis that point the same results (1).

Line22: not really to be included in the paper, but at time of herpes zoster (HZ) onset VZV DNA has also been found in broncho alveolar lavages (2,3), urine (personal unpublished results on immunosuppressed) and cerebrospinal fluids (4).

You quote Guiraud and Quinlivan, which found a high proportion of positive patients with VZV DNA in blood, and oppose it to Park. Of note, Park used Plasma instead of blood. As VZV is thought to be bound to cells in blood, levels in plasma are almost half that in blood, and as low as 5% (5), which can explain a lower positivity since viral loads are frequently close to the limit of detection. Moreover DNA extraction method and PCR were different between the studies, which could partially explain the results.

Line 34: can you specify if you use whole blood, plasma or serum?

Line 36: Which extraction method did you use? What was the limit of detection of your assay? (the affiliation is the same for the last author so I guess each part of the process are identical).

Out of curiosity, have you tried to process samples with either a simplex commercial assay or the simplex part of your assay? It might help to strengthen your results and ease further inter publication comparison. Moreover, in the initial publication (6) there was no saliva included to validate the assay on this specific matrix. Has this been performed since?

I have several questions regarding the patients included, most of them align with the strobe guidelines (7):

- Was this a retrospective or prospective study? (I guess it was a prospective one since you managed to get saliva from all your patients but it's not detailed in the manuscript).
- You almost don't describe the patients. How many of them were immunosuppressed in each group? If so, which kind of immunosuppression? What were the symptoms of the atypical herpes zoster group? Who adjudicated the diagnoses?
- Were there any differences between patient' characteristics of the positive and negative saliva subgroup of the atypical and typical herpes zoster?
- For the atypical HZ, was there any follow-up? Did any of them developed vesicles? Were there treated with antiviral agents?
- What was time from onset of HZ?
- How were recruited the healthy controls?

- How did you determine the number of patients to include? Were they consecutive or you selected them based on samples available?

For saliva, do you have a quantification (copy/mL of saliva or per million cell ?) It would be interesting to have a level of replication and see if the median level is close to the limit of detection.

Line 43 “suggesting the blood might not be the best samples for diagnosis of VZV infection”. In our laboratory, VZV levels are often higher than the limit of quantification in vesicles (if you have this information for your samples, you might add it in the table) and systematically positive. Moreover, it allows a non-invasive and easy diagnosis so I guess you could be a bit more assertive. Finally, if you compare positive rates of blister fluid and saliva and blister fluid and blood you would have a p-value = 0.0002069 for the first and p-value = 2.485e-12 for the later (using prop.test in R with continuity correction).

References:

1. Chen N, Li Q, Yang J, Zhou M, Zhou D, He L. Antiviral treatment for preventing postherpetic neuralgia. Cochrane Neuromuscular Group, éditeur. Cochrane Database of Systematic Reviews [Internet]. 6 févr 2014 [cité 14 juill 2024];2014(2). Disponible sur: <http://doi.wiley.com/10.1002/14651858.CD006866.pub3>
2. Guiraud V, Burrel S, Luyt CE, Boutolleau D. Prevalence and clinical relevance of VZV lung detection in intensive care unit: A retrospective cohort study. *Journal of Clinical Virology*. juill 2023;164:105470.
3. Engelmann I, Petzold DR, Kosinska A, Hepkema BG, Schulz TF, Heim A. Rapid quantitative PCR assays for the simultaneous detection of herpes simplex virus, varicella zoster virus, cytomegalovirus, Epstein-Barr virus, and human herpesvirus 6 DNA in blood and other clinical specimens. *Journal of Medical Virology*. mars 2008;80(3):467-77.
4. Haanpaa M, Dastidar P, Weinberg A, Levin M, Miettinen A, Lapinlampi A, et al. CSF and MRI findings in patients with acute herpes zoster. *Neurology*. 1 nov 1998;51(5):1405-11.
5. De Jong MD, Weel JFL, Schuurman T, Wertheim-van Dillen PME, Boom R. Quantitation of Varicella-Zoster Virus DNA in Whole Blood, Plasma, and Serum by PCR and Electrochemiluminescence. *J Clin Microbiol*. juill 2000;38(7):2568-73.
6. Li Y, Wan Z, Zuo L, Li S, Liu H, Ma Y, et al. A Novel 2-dimensional Multiplex qPCR Assay for Single-Tube Detection of Nine Human Herpesviruses. *Virology*. août 2021;36(4):746-54.
7. Von Elm E, Altman DG, Egger M, Pocock SJ, Gøtzsche PC, Vandenbroucke JP. The Strengthening the Reporting of Observational Studies in Epidemiology (STROBE) statement: guidelines for reporting observational studies. *The Lancet*. oct 2007;370(9596):1453-7.

Response to Reviewers' Comments

Journal: *Microbiology Spectrum*

Manuscript ID: Spectrum01549-24

Title: Detection of varicella-zoster virus in blood and saliva of patients with typical and atypical herpes zoster

Authors: Yinling Li¹, Xiuli Zhao¹, Jiewei Chen¹, Zhen Pu¹, Chiyu Zhang², Aizhen Zhou^{1,*}, Zhenzhou Wan^{1,*}

Response to Editor's Comments

I agree with the reviewers' comments about providing clinical information, particularly whether patients were immunocompromised. Please see attachment for comments of Reviewer 2.

Response: We thank you and the reviewers for the valuable suggestions and comments. In our study, two patients might be immunocompromised since they had received surgery to remove cancer with and without subsequent chemotherapy about two years before sample collection in this study. Some necessary information required by the reviewers have been added in the revised manuscript.

Response to Reviewers' Comments

Reviewer #1: In the small methods paragraph please specify statistical methods applied.

Response: As suggested, we added the statistical methods in the revised manuscript: "Comparisons were performed using the t-test or chi-square test with SPSS 26.0" (line 139).

Lack of description on how non vesicular zoster was diagnosed and what specifics the patients had, especially how the difference from HSV was made.

Response: Patients with atypical herpes zoster were diagnosed according to clinical symptoms and further imaging examination. "The atypical HZ contains incomplete herpes zoster, zoster sine herpete and disseminated zoster" (lines 44-45) In addition, we mentioned "because of the similarities in clinical presentations of VZV infection with herpes simplex virus infection, as well as other diseases (e.g. erythema multiforme)" (lines 22-24) in the revised manuscript.

Reviewer #2:

Overall comment: The paper is clear, well-written and easy to read. I hope it will get published. Results from blood positivity are a bit surprising from my experience but could be explained by different patient populations and different diagnosis methods.

Response: We thank you for your valuable comments and suggestions. We have carefully addressed your concerns.

Line 15-16: I don't agree with the assumption that early treatment decrease PHN. The reference 2 only addresses median time to resolution of the acute symptoms, not PHN. Their figure 1 for instance is extremely relevant: median time to resolution is shorter but proportion of patients with ZAP at 24 weeks is roughly similar. There is also a Cochran meta-analysis that point the same results (1).

Response: We thank you for pointing out this inappropriate description. The sentence has been revised as “Antiviral therapy can significantly decrease the duration and severity of HZ-associated pain”. (lines 18-19)

Line22: not really to be included in the paper, but at time of herpes zoster (HZ) onset VZV DNA has also been found in broncho alveolar lavages (2,3), urine (personal unpublished results on immunosuppressed) and cerebrospinal fluids (4).

Response: We thank you for pointing out this inappropriate description. The sentence has been revised as “VZV DNA can be detected in blood, saliva, broncho alveolar lavages, cerebrospinal fluids and even urine at the time of HZ onset”. (lines 26-27).

You quote Guiraud and Quinlivan, which found a high proportion of positive patients with VZV DNA in blood, and oppose it to Park. Of note, Park used Plasma instead of blood. As VZV is thought to be bound to cells in blood, levels in plasma are almost half that in blood, and as low as 5% (5), which can explain a lower positivity since viral loads are frequently close to the limit of detection. Moreover DNA extraction method and PCR were different between the studies, which could partially explain the results.

Response: We thank you for pointing out this crucial point. This section has been revised as “In addition, the detection rate of salivary VZV DNA in this study was relatively lower than those in the previous studies [12], which might be partly ascribed to different methodologies (including DNA extraction and the qPCR assay)”. (lines 59-61)

Line 34: can you specify if you use whole blood, plasma or serum?

Response: We used plasma.

Line 36: Which extraction method did you use? What was the limit of detection of your assay? (the affiliation is the same for the last author so I guess each part of the process are identical).

Response: Viral DNA was extracted using a qEx-DNA/RNA viral extraction kit (Tianlong, Xi'an, China). This information was added in the revised manuscript. In addition, the qPCR assay used in this study was previously developed by ourselves and the limit of detection was 30 copies per 25 μ L reaction.

Out of curiosity, have you tried to process samples with either a simplex commercial assay or the simplex part of your assay? It might help to strengthen your results and ease further inter publication comparison. Moreover, in the initial publication (6) there was no saliva included to validate the assay on this specific matrix. Has this been performed since?

Response: We agree with your comments. In this study, we used a simplex qPCR assay for VZV DNA detection, and no commercial assay was used. In addition, in our previous paper for the development of the multiplex qPCR assay, we used whole-blood and vesicular (rather than saliva) fluid samples for evaluation. Since the development of the assay, no saliva samples were used to assess the assay. However, according to our experience in developments of various nucleic acid amplification methods (e.g. HiFi-LAMP, and mismatch-tolerant qPCR) (*ACS Sensors*, 2022, 7:730-739; *Microbiol Spectrum*. 2024, 12(4): e0413323; *Anal Bioanal Chem*. 2024, 416(8):1971-1982; *Diagnostics*, 2023, 13: 1530; *Front Microbiol* 2019, 10:1056; *BioTechniques* 2019, 66:225-230), the method used in this study is believed to

have a good performance for detection of VZV DNA in saliva. However, because of no direct available data to support this assumption, we mentioned it as a limitation of this study in the revised manuscript.

I have several questions regarding the patients included, most of them align with the strobe guidelines (7):

- Was this a retrospective or prospective study? (I guess it was a prospective one since you managed to get saliva from all your patients but it's not detailed in the manuscript).

Response: Yes, this was a prospective study.

- You almost don't describe the patients. How many of them were immunosuppressed in each group? If so, which kind of immunosuppression? What were the symptoms of the atypical herpes zoster group? Who adjudicated the diagnoses?

Response: In our study, two patients might be immunocompromised since they had received surgery to remove cancer with and without subsequent chemotherapy about two years before sample collection in this study. None of the patients with typical herpes zoster were immunosuppressed. The symptoms of the atypical herpes zoster group include zonally distributed erythema and papules with pain, or no rash but having paroxysmal, pins-and-needles pain, or disseminated herpes zoster. All patients enrolled in this study were diagnosed by the dermatologists (Authors: Yinling Li, Jiewei Chen, Zhen Pu, and Aizhen Zhou) of Taizhou Fourth People's Hospital.

The information above-mentioned have been added in the revised manuscript.

- Were there any differences between patient' characteristics of the positive and negative saliva subgroup of the atypical and typical herpes zoster?

Response: No differences were found in clinical characteristics between the VZV DNA positive and negative saliva subgroups of regardless of the atypical and typical HZ groups.

- For the atypical HZ, was there any follow-up? Did any of them developed vesicles? Were there treated with antiviral agents?

Response: The atypical HZ patients were not followed up. So, we did not sure whether or how many of them developed vesicles. Of 35 atypical HZ patients, 33 received antiviral treatment, including three initiating antiviral treatment before enrollment (sampling). Two patients did not receive antiviral treatment.

All typical HZ patients received antiviral treatment.

- What was time from onset of HZ?

Response: Thank you for this suggestion. We analyzed and added the data on the times from onset of both the atypical and typical HZ groups in the revised manuscript (Please see the Table below). The mean time from onset was 5.1 and 4.1 days for atypical and typical HZ groups, respectively. The VZV DNA positive atypical HZ patients had shorter times from onset than negative patients.

Table 1. The times from onset of both the atypical and typical HZ groups.

Groups	VZV DNA positive: Duration (scope)	VZV DNA negative: Duration (scope)	P value	Total: Duration (scope)	P value
Atypical HZ (Non-vesicular)	3.6 days (2-7)	5.5 days (2-10)	0.044	5.1 days (2-10)	0.08

Typical HZ (Vesicular)	4.8 days (1-10)	3.7 days (1-7)	0.214	4.1 days (1-10)	
-----------------	----------------	-------	-----------------	--

The data are shown in mean time from symptom onset with a scope.

- How were recruited the healthy controls?

Response: The staff at Taizhou Fourth People's Hospital who have no major underlying diseases were recruited as healthy controls.

- How did you determine the number of patients to include? Were they consecutive or you selected them based on samples available?

Response: The number of patients in this study was mainly based on sample availability.

For saliva, do you have a quantification (copy/mL of saliva or per million cell?) It would be interesting to have a level of replication and see if the median level is close to the limit of detection.

Response: Unfortunately, we did not perform parallelly amplification using DNA standards of different dilutions. So, we are unable to obtain the viral load of VZV DNA in the samples. As an alternative, we compared the Ct values (linearly negative correlation with log₁₀ viral copy number) between different groups.

Line 43 "suggesting the blood might not be the best samples for diagnosis of VZV infection". In our laboratory, VZV levels are often higher than the limit of quantification in vesicles (if you have this information for your samples, you might add it in the table) and systematically positive. Moreover, it allows a non-invasive and easy diagnosis so I guess you could be a bit more assertive. Finally, if you compare positive rates of blister fluid and saliva and blister fluid and blood you would have a p-value = 0.0002069 for the first and p-value = 2.485e-12 for the later (using prop.test in R with continuity correction).

Response: We thank you very much for these comments and suggestions that are really helpful for the improvement of the paper. We added and mentioned the data and emphasize the advantage of saliva in diagnosis in the revised manuscript. (lines 63-67)

Reviewer #3: In this paper, Li et al detected and compared the VZV DNA among blood and saliva samples from patients with typical and atypical herpes zoster. The results have clinical implication for early diagnosis of atypical herpes zoster, as well as distinguishing VZV infection from HSV infection, and deserve publication.

Response: We thank you for the positive comments.

Comments:

1.The definition of atypical herpes zoster is not completely accurate. Atypical herpes zoster also contains disseminated herpes zoster and visceral herpes zoster, etc. Furthermore, some cases without skin lesions have pain.

Response: Thank you for the comments and suggestions. We have updated the definition of atypical herpes zoster in the revised manuscript. (lines 44-45 please also see the response to the comment 2 of the reviewer #1)

2.The clinical implication and/or importance of accurate diagnosis of VZV infection should be discussed. For example, how to distinguish atypical herpes zoster from other visceral diseases, and to distinguish typical and atypical herpes zoster from herpes simplex when it occurs in the skin and mucosa.

Response: Thank you for this suggestion. We discussed the clinical implication and importance of accurate

diagnosis of VZV infection in the revised manuscript. (lines 19-21 and 24-25) By the way, it may be relatively easy to clinically distinguish HZ from HSV infection since both diseases seem to have symptoms in different body sites. (Please also see the response to the comment 2 of the reviewer #1).

3. More clinical information of the patients should be provided. For example, did the patients receive antiviral treatment before sample collection?

Response: As suggested, the information was added in the revised manuscript.

4. The viral load or at least the CT values of the qPCR assay can be analyzed and compared between different groups.

Response: As suggested, we added and compared the data of the Ct values of the qPCR assay between different groups in the revised manuscript. (lines 63-67 and Fig. 1).

Re: Spectrum01549-24R1 (Detection of varicella-zoster virus in blood and saliva of patients with typical and atypical herpes zoster)

Dear Dr. Zhenzhou Wan:

Thank you for the privilege of reviewing your work. Below you will find my comments, instructions from the Spectrum editorial office, and the reviewer comments.

Please include the following information in the revised manuscript:

1. Explain that the study was prospective.
2. Indicate the number of immunocompetent and immunosuppressed patients. This can be included in the letter or in a footnote under the table. On a separate note, I do not consider remote cancer chemotherapy an immunocompromising condition.
3. Include a statement that staff volunteers gave their samples freely and without duress (This was reviewed by our Ethics team).
4. Was VZV DNA detected in the saliva of healthy controls? Please explain.

Revision Guidelines

Sincerely,
Paschalis Vergidis
Editor
Microbiology Spectrum

Response to Reviewers' Comments

Journal: *Microbiology Spectrum*

Manuscript ID: Spectrum01549-24R1

Title: Detection of varicella-zoster virus in saliva and plasma of patients with typical and atypical herpes zoster

Authors: Yinling Li, Xiuli Zhao, Jiewei Chen, Zhen Pu, Chiyu Zhang, Aizhen Zhou,* , Zhenzhou Wan*

Response to the Editor's comments

1.Explain that the study was prospective.

Response: As suggested, we mentioned this study is a prospective study.

"Here, we performed a prospective study to detected and compared the presence of VZV DNA in plasma" (line 39).

2.Indicate the number of immunocompetent and immunosuppressed patients. This can be included in the letter or in a footnote under the table. On a separate note, I do not consider remote cancer chemotherapy an immunocompromising condition.

Response: As suggested, patients previously receiving cancer chemotherapy were not considered an immunocompromising condition. Therefore, all 70 patients were immunocompetent. The information was added in the revised manuscript as a footnote of Table 1:

"All patients were immunocompetent at least during this study. In addition, there were two patients who had received surgery to remove cancer with and without subsequent chemotherapy about two years before sample collection."(lines 147-149).

3.Include a statement that staff volunteers gave their samples freely and without duress (This was reviewed by our Ethics team).

Response: Done. (line 49).

4. Was VZV DNA detected in the saliva of healthy controls? Please explain.

Response: We previously mentioned that No VZV DNA was detected in saliva of healthy people (line 55). Now, we moved this sentence forward to line 51.

Re: Spectrum01549-24R2 (Detection of varicella-zoster virus in saliva and plasma of patients with typical and atypical herpes zoster)

Dear Dr. Zhenzhou Wan:

Your manuscript has been accepted, and I am forwarding it to the ASM production staff for publication. Your paper will first be checked to make sure all elements meet the technical requirements. ASM staff will contact you if anything needs to be revised before copyediting and production can begin. Otherwise, you will be notified when your proofs are ready to be viewed.

Sincerely,
Paschalis Vergidis
Editor
Microbiology Spectrum